# Different Approach to Horses—The Use of Equid Remains in the Early Middle Ages on the Example of Ostrów Tumski in Wroclaw

**DOI:** 10.3390/ani10122294

**Published:** 2020-12-04

**Authors:** Krzysztof Jaworski, Aleksandra Pankiewicz, Aleksander Chrószcz, Dominik Poradowski

**Affiliations:** 1Institute of Archaeology, University of Wroclaw, Szewska 48, 50-139 Wrocław, Poland; krzysztof.jaworski@uwr.edu.pl; 2Department of Biostructure and Animal Physiology, Faculty of Veterinary Medicine, Wroclaw University of Environmental and Life Sciences, Kożuchowska 1/3, 51-631 Wrocław, Poland; aleksander.chroszcz@upwr.edu.pl (A.C.); dominik.poradowski@upwr.edu.pl (D.P.)

**Keywords:** horse, use of remains, archeozoology, early middle ages, production in strongholds

## Abstract

**Simple Summary:**

Wroclaw, the capital of Silesia located in south-western Poland, was one of the most important settlement agglomerations in the Western Slavic region in the period from the mid of the 10th to the 12th/13th century. The center of Wroclaw was a multi-part fortified settlement located on the Ostrów Tumski Island, the seat of the ducal and church authorities in the Middle Ages. Apart from spectacular finds of architectural monuments and elite products from various spheres of material culture, thousands of seemingly less interesting artefacts were found in Ostrów Tumski, among them a series of over 100,000 animal remains. A detailed and multifaceted analysis of the archaeozoological materials from Wroclaw clearly shows various interactions between the inhabitants of the stronghold and the animals living among them or nearby. One of the most important species in everyday life of the settlement was the horse. This animal is primarily associated with horseback riding and knighthood, possibly also with draught purposes. However, were these really the only functions of the horse and other equids? The aim of this article is to show the varied role of the horse, inter alia, its usage as a raw material source for the production of items made of bone, hair and leather, and sometimes even for food.

**Abstract:**

The following article concerns the functional use of horse bones in the early Middle Ages (mainly in the period from the mid of the 10th to the 12th/13th century). The authors try to explain how such remains were used and how common it was. It is also discussed whether the special role of the horse in medieval societies somehow restricted its post-mortem usage, or perhaps there was no difference between the skeletal remains of horses and other species in this regard. For this purpose, statistical calculations on the use of the bones of various mammals were made. Only the remains of the species determined during the archaeozoological analysis were taken into account. The specific use of individual parts of a horse skeleton was also noted. In addition, the analysis also encompasses all other types of horse remains that could be used by humans (hide, hair, etc.). The consumption of horse meat was discussed separately: on the basis of the preserved traces, an attempt was made to determine whether it had happened, and if so, how popular it had been. Overall, such comprehensive analysis aims to show the various roles of the horse. It was not only a mount, but also a beast of burden, a source of food and raw material as well. The main purpose of this study is to describe the role of horses in human medieval societies of Ostrów Tumski on the basis of accessible equid remains. The highlighting of the human–horse relationship in the past allows us to understand the importance and value of the horse both as a life companion and the source of food or leather and bone tools.

## 1. Introduction

The horse played a special role in early medieval culture (for Central and Eastern Europe, the 6th to the 12th/13th century). It was a symbolic animal, an inseparable element of the sacred, not only for the Slavs, but also for their neighbours—Germanic, Baltic and nomadic peoples. It accompanied a man both in his life and in the afterlife. That is best exemplified by the burying of warriors with their horses, e.g., in nomads or the Balts, or their presence in the rituals of the Slavs (e.g., fortune-telling, sacral offerings, figural sculptures and other images). The horse is also often associated with elites, not only as a means of transport and an element of military equipment, but also as a certain indicator of social prestige. It is also associated with this role in the subject literature [1,2,3,4]. Medieval horses can be classified into three types: destriers, coursers, and rounceys. Usually the body size of a horse was moderate, rarely higher at the withers than 157 cm, because larger horses were more difficult to maintain and less adaptable from a logistical and territorial point of view. Early medieval destriers were larger than the two other types, and were useful for heavy armoured knights. However, they were still smaller than draught animals.

In archaeozoological practice, the most common study materials are the post-consumption remains of domestic animals, which constitute the overwhelming majority of bone remains with determined species and anatomical affiliation [5,6,7,8]. The bone remains of horses usually do not exceed 4% of NISP(the number of identified specimens), and in the distribution of anatomical elements a deficiency of skeletal elements of limb distal parts is observed, which is often interpreted as a result of their acquisition as a raw material. Much less often the bone remains are found in an anatomical order (articulated skeletons), which is mainly related to ritual burials [9], or the decomposition of the animal carcass on the so-called knacker’s yards [10]. All kinds of pathological changes associated with diseases that give visible changes in the bone tissue [11] are very interesting for the researchers. In the case of horses, they most often result from the improper use of animals and mainly concern the musculoskeletal system [12,13].

An additional problem of archaeozoological studies is the differentiation of horses’ and other equids’ skeletal remains. The good and critical review of the accessible methods invented for achieving this aim was published by Baxter [14]. Except from the well-known Mesolithic–Neolithic presence of wild equids in Europe (including the Baltic region), the domestication of horses in the Ukrainian steppes and the presence of other wild equids from the Pleistocene [14,15,16], we still have problems with the successful identification of skeletal remains. The earliest written sources mentioning horses in medieval Poland come from The Chronicles of Gallus Anonymus and the descriptions of a Jewish traveller named Abraham ben Jacob (966 AD), i.e., ‘a country where the weather is healthy, the earth is fertile, the forest supplies honey, waters give fish, the knights are warlike, peasants are laborious, horses are strong, oxes are willing to plough, cows give milk, sheep supply wool’ [17]. Other equids’ existences in Poland are rarely archaeologically confirmed at archaeological sites dated back to the Roman influences period [18]. In general, the unearthed equids’ skeletal remains were not numerous and strongly fragmented, causing the above-mentioned identification problems. Therefore, the term ‘horse’ must be understood widely in this paper, as corresponding to a whole equid family.

Finally, the horse meat eating taboo seems to be a recurrent phenomenon in human history. The earliest signs of equid butchery was proven in North Africa 1.78 million years ago [18]. Probably, the hypophagia was not only the Scythes’, way and not in starving periods or wars only [14]. In the Middle Ages, in 732 AD, Pope Gregory III began intense efforts to stop the ritual consumption of horse meat in pagan practices [19]. Simultaneously, pagan Slavs used to treat horses as sacred animals (Wolin and Brenna), which influenced their horse–human relations and horse value [20]. Medieval horse meat consumption was rare, not only due to religious prohibitions, but also due to the animal’s value (price) and its role in a human life (i.e., a warhorse or a hunting horse) [1,3].

Today, the frequency of horse meat consumption varies in European countries; the taboo-like prejudices are still alive, but simultaneously the horse meat consumption has survived among historically nomadic peoples such as the Tatars, the Yakuts, the Kyrgyz, and the Kazakhs, as the relic of Neolithic hunting for wild-living equids [21].

However, the aim of this article is to present a completely different role of the horse, which was also a functional source of various raw materials. The practical application was found mainly for bones, less often for hide and hair. The authors try to show whether there is any regularity regarding the preference of the raw material used for the production of specific items and whether this had changed over time. Finally, a horse was also a source of meat. Its consumption in the past is unquestionable, and often studied by the researchers [22,23,24]. An attempt is made to determine the scale of this phenomenon in the early Middle Ages, and how the traces of such practices look.

## 2. Materials and Methods

The studied site of Ostrów Tumski in Wroclaw used to be one of the most important centers in Poland in the period from the middle of the 10th to the 12th/13th centuries. It is also one of the best archaeologically recognized sites in this part of Central Europe. During many years of research, numerous horse bones were discovered there, as well as other remains of the use of these animals.

All traces related to the use of the horse-based raw materials, i.e., processed bones, hides and hair, as well as debris that may be associated with consumption, were taken into account. These observations will be compared with the frequency of similar traces found on the remains of other animal species (mainly the most numerous bones of pigs and cattle). All identified horse bones were submitted to standard archaeozoological analysis, without differentiation between horses and other equids. The frequency of the occurrence of species and the distribution of anatomical elements was estimated on the basis of the total number of bone fragments (TNF), the number of identified specimens (NISP) and the normal distribution of anatomical elements in a horse skeleton. Both the results obtained in recent years [6,7,8,25,26,27,28] and the archival analyses [29,30,31] were used. The osteometric investigation was carried out with standard archaeozoological techniques. All measurements were carried out three times in order to estimate the mean value. The height at the withers was computed using the craniometry and the metapodial bones’ morphometry. The animal age was estimated on the basis of the animal’s dentition and the long bones’ epiphyseal fusion status. Moreover, the following features were taken into account: degenerative traces on the bones indicating a beast of burden (draught animals), and traces related to bone processing and horse meat consumption.

During the archaeological research conducted so far at Ostrów Tumski, over 100 items made of horse bones have been identified (about 12% of all bone and horn products). They were found in the wide-area trenches in the eastern, larger part of the stronghold, in groups of several dozen artefacts [25,26,27,28]. They are less common in the western part of the complex (*castrum minus*).

Statistics constitute another important element of the analysis. An attempt is made to determine the real use of horse bones as raw materials by comparing the number of products made of the given raw material per 100 unprocessed items of the same material. Establishing a fixed point of reference allows its direct comparison for different types of bones. Calculating the usage of a specific bone is based on a simple formula: y/z = 100/x, where ‘y’ is the number of excavated, unprocessed remains of the species, ‘z’ is the number of artefacts made of this material, and ‘x’ is the usage rate. For example, if there are 8 artefacts made of horse bone out of 40 unprocessed horse bones, the usage rate would be 100:20 [25,26]. This method, consistently used also for various skeletal parts of other animals, allows an objective recognition of the preferences and the scale of obtaining bone material by those who processed it.

When analyzing the usage of horse raw material, hides were also considered. They were found in the majority of trenches at Ostrów Tumski [32]. On the other hand, the presence of hair and fur was less useful for the purposes of the study, as the researchers did not always determine to which species they belonged. Here, analogies from other sites were used, which also illustrated other phenomena related to the use of horse-based raw materials. The use of these methods allows us to determine the role of a horse among other farm animals, as well as to estimate the physical condition of individual specimens, the remains of which were discovered at the site.

## 3. Results and Discussion

### 3.1. Statistical Analysis of the Horse Skeletal Remains

A clear disproportion was observed in the occurrence of horse skeletal remains between individual parts of the stronghold in Ostrów Tumski. The low amount of horse bone products in the smaller, western part of the complex is most likely due to the fact that it has not been thoroughly excavated yet, not a reflection of the lack of interest in such material by the stronghold inhabitants (Figure 1). Such a distribution of the horse bone remains and a significant concentration of them in the eastern and central parts of the stronghold (*castrum maius*) is probably also not the result of the existence of a bone and antler workshop in this area. Processed animal bones were found in all excavations. This accumulation is rather the result of a higher frequency of finds (including bones) in this zone. This is also confirmed by the stable percentages of horse bone products in relation to all bone and horn objects found around the island and in the stronghold, ranging from 9 to 17.5% (usually around 12%). Such high frequency of horse bones used as raw materials in handicraft production can be compared with the low amount of unprocessed horse bones in the number of identified specimens (NISP) from Ostrów Tumski in Wroclaw (Table 1)—ranging from 1.31% to 3.7% [6,7,25,26,27,28,29,30,31].

The percentage of horse bones found in some sites is even higher. For example, in the Moravian Mikulčice, it amounts to as much as 32.5% of all bone and horn artefacts [33]. The opposite is the well-known Ostrówek site in Opole (about 100 km away from Wroclaw), where a long series of bone and horn products was obtained (469 artefacts), but the amount of horse bone items was much lower: only 4.3% (20 artefacts) [26,34]. Such a difference (nearly three times more items of horse bones in Wroclaw) is most likely caused by some local, individual features of both sites.

The horse bone products were excavated in layers from different periods of the settlement activity, but most of them were found in older strata. In total, 63% of them were located in the central part of the stronghold (trenches I-II, III and IIID) in layers dated from the mid-10th to the second half of the 11th century. More than 70% of the unprocessed horse remains were also excavated from these layers. Only in one of the excavations (IIIF) were these proportions distorted. More finished products and unprocessed horse bones were found here in younger stratigraphic contexts (the second half of the 11th to the second half of the 12th century) [7,26]. It was also observed that the number of horse bone products within a given layer was proportional to the number of unprocessed animal bones present in it.

The percentage of horse bones decreases in the youngest stronghold layers dating back to the first half of the 13th century. There may be several explanations for that phenomenon. As a chartered town started developing on the other side of the Odra River, the stronghold in Ostrów Tumski started losing its importance. Social transformations were accompanied by economic changes, including the way of using bone material. Bovine-based products started gaining popularity, while others, including horse bone skates, were disappearing [27,35]. Perhaps the phenomenon of the diversification of pork in favor of beef in cities chartered under the Magdeburg Law, observed in the changing share of cattle and pig skeletal remains, was not limited to these species [8], but also to some extent influenced the consumption of horse meat or the using of their skeletal elements in crafts.

### 3.2. Use of Horse Bones: Raw Material Preferences

Horse bone products were made almost exclusively from the remains of an animal appendicular skeleton. The most frequently processed were the metacarpal and metatarsal bones (40 products from the former, 19 from the latter, 3 from one or the other) and the radial bones (29 items) (Figure 2). It should be assumed that these particular bones, perfectly suited for the production of bone skates and skids, were carefully collected by their manufacturers (Figure 3a–e). This is indicated by the frequent usage of these bones, visible in excavations, ranging for metacarpal bones from 100:28.12 to as much as 100:91.66 (in trench III), for metatarsal bones from 100:27.78 to 100:40 and 100:45, and for radial bones from 100:45 to 100:50. The metacarpal and metatarsal bones were also the basic raw materials for the production of skates and skids in other early medieval European sites [33,34,36,37,38,39,40]. Bovine bones were used less frequently for this purpose [26,34,36], as their higher amount is noted only on single sites, among others in the trade emporium in Truso, located in the Balts lands [41]. The overwhelming advantage of bovine bones in this site may possibly be related to the special role of horses in the Balts societies. However, there are also other products made of horse bones (e.g., pawns for board games), which undermines the argument about reluctance to use horse raw material. The use of bovine bones may be associated with the time of existence of the emporium in Truso (mainly the beginning of the 9th to the end of the 10th century), when horse bones were not so often used for the production of skates in Polish lands.

Skates or skids made of the horse tibia are sporadic [26]. An animal mandible was also used in making small sledges [36]. The morphological features suggest the more frequent use of the horse’s metapodium and radius bones. They are long, with epiphyses ossifying respectively at around 15 months and between 15 and 41 months of age, showing quite high bilateral symmetry and strength. As for the mandible, unlike in cattle (lack of synosteosis), the symphysis of the horse mandible ossifies, creating a permanent bone fusion [42].

Processed horse mandibles could also be used as musical instruments, such as idiophones [39]. Horse teeth were also used as pawns in board games [41]. Due to their irregular cuboid shape and cross-sectional structure, the long-crowned (hypselodontic) cheek teeth of a horse seem to be an excellent material for this purpose [42]. Amulets were also made of horse bone, although this was not common [34]. At Ostrów Tumski, the presence of a comb made from the capsule of a horse’s hoof was also confirmed (Figure 3k). It was discovered in a layer from the second half of the 11th century. It is the only example of such usage of this material in Poland. In another case, a horse rib was used for a carefully crafted knife holder (Figure 3f). Although this is an isolated artefact, horse bones were known to be used for such purposes until the beginning of the 20th century [43]. However, the mentioned artefacts rarely appear in the early Middle Ages (Figure 2 and Figure 3), and the use of the bones they were made from was of marginal importance.

The so-called spikes were also created from horse skeletal remains. They were universal items that could be used for piercing, knitting, engraving, and many other things. The raw material used for making them was also varied, including horse’s splint bones (the second and fourth metacarpal or metatarsal bones), although such items were not common in the studied collection [25,26,27,28,38]. Eight such artefacts were identified in Ostrów Tumski (Figure 3g–i), with the usage rate ranging from 100:4.55 to 100:5.17 [26]. It is possible that splint bones were obtained while processing the nearby metapodia (Figure 2 and Figure 3). Therefore, among the parts of a horse skeleton, long and straight bones were preferred for tools (Figure 2). This is consistent with the results of studies from other sites [25,26,28,29,30,31,32,33,34,36,37,38,39,40].

The demand for long horse bones was still high in the late Middle Ages. It is evidenced by the discovery of four almost complete horse skeletons in one of the late medieval wells. Larger bones, less susceptible to destruction, were preserved in all four specimens. The metacarpal and metatarsal bones were less complete, as only one of the former and two of the latter were discovered, instead of a total number of eight. Radial bones were preserved more often (six out of eight), as their processing was probably abandoned in the early Middle Ages [27,35]. The demand for raw material used to produce certain items, including ice skates, is also evidenced by the specific distribution of horse bones in the excavations. The unprocessed skeletal fragments come mainly from limbs, skulls and adjacent parts of the body (cervical vertebrae) [7,30]. The presence of limb bones can be explained by the collection of raw material (Scheme 1). Skulls, on the other hand, especially those found within households, were most likely associated with magical or ritual practices (foundation deposits) [4,40].

The comparison of horse bone products found in the excavations and the number of their unprocessed remains also confirms the popularity of that material among the past Wroclaw inhabitants. As was mentioned earlier, horse bone and horn products constituted respectively 9.3% and 12.7% of all such artefacts in the excavations located in the central part of Ostrów Tumski, while the percentage of unprocessed horse remains in the same excavations was relatively much smaller—from 1.31% to 3.7% of the NISP (Table 1).

### 3.3. Determining the Features of Wroclaw Horses Based on the Analysis of the Remains and Artefacts

The dimensions of bone items strictly depend on the size of the specimen (height at the withers) from which the raw material was obtained [44]. Based on bone material from the stronghold in Wroclaw, W. Chrzanowska [29] distinguished four types of horse sizes:(a)small horses, 124.0–128.0 cm at the withers (15.5% of total number);(b)small horses among medium ones, 132.2–136.0 cm at the withers (31.1%);(c)medium horses, 136.4–144.0 cm at the withers (42.2%);(d)large horses among medium ones, 144.7–152.0 cm at the withers (11.1%).

The most numerous were the remains of the horses from the size group (c), which coincides with the findings of other researchers [7,45] and is also similar to the measurement results obtained from other important settlement sites [46,47,48]. The analysis of horse bone products, mainly skates made of long bones, showed that they came from animals belonging to all size types. However, the items made of smaller animal bones (groups a and b) prevailed. It is puzzling and so far unexplained why the horses from the most numerous group in the stronghold (c) provided raw material for the production of only two artefacts. In general, the number of horses calculated on the basis of horse-based artefacts differs significantly from that based on unprocessed bones. The shape of long bones, apart from the height of the animal, is also influenced by its morphological type. Perhaps long bones from smaller animals were more robust than those obtained from bigger ones, being similar in structure to the skeletons of draught horses, and thus being more suitable for processing [42].

The question is whether the shortage of skates from the bones of bigger specimens is the result of a selective choice of raw materials for the production of skates, or because of a reluctance to use the bones of animals higher at the withers (and thus probably more valuable). Such a phenomenon was common not only in Wroclaw, but was also observed in artefacts from Poznan [46]. It is worth noting that in both of the sites (Wroclaw and Poznan) we are dealing with strongholds of the highest rank in the territory of Poland, partially inhabited by the knightage. Perhaps larger horses were bred for the needs of this social group, probably for fighting. War horses were considered extremely valuable. Horses of various breeds, specially trained for this purpose, were also used during armed conflicts [1,3]. Horses were also a valuable commodity in Central Europe, as evidenced by the 10th century information on trade in these animals at the Prague market [40].

The phenomena are hard to find in the archaeological material, an example of which is the analysis of the chronology of the occurrence of remains from horses of various sizes. There was no correlation between the horses’ size and the chronology of layers in Ostrów Tumski. Artefacts made of horse bones of different sizes were often found in the same layers. At the same time, the bone products made of horses from group (d) were previously found in older settlement levels. The presence of horse bones of considerable size in the strata from the 10th and 11th centuries seems to confirm the observations resulting from the analysis of Wroclaw horseshoes. It concluded that bigger horses with large hooves were occasionally bred already in the 10th–11th centuries, apart from shorter horses with small and medium hooves [29,48]. Moreover, their number clearly increased in the 12th century. [49] However, it should be noted that the hoof size is influenced not only by the height at the withers, but also by the horse breed (warmbloods vs. cold-blooded draught horses).

Studies on the age structure of horses from the Wroclaw stronghold showed that they did not die young, and some even lived to see very old age. The most numerous lived 7–8 years; two were 17–18 years old at the time of death and one horse survived only 1.5 year [30,31]. Although such results differ slightly from the data obtained in other early medieval sites, the general trends are similar (small number of foals, the majority of animals were middle-aged, several older ones) [40,41]. This clearly shows that horses were kept mainly for transportation purposes. This is confirmed by the traces on horse remains in Wroclaw, visible in the preserved elements of the locomotor system, indicating overloads resulting from using the animals as draught animals [7,8].

### 3.4. Use of Other Horse-Based Raw Material

Contrary to bone, the processing of horse hide was of marginal importance. Among the 2626 hides found in the Wroclaw stronghold, only 26 fragments were identified as horse leather. Thus, they constituted only 0.99% of all the material. They were concentrated (21 out of 26 fragments) within the layers of one of the trenches (III) in the central part of the stronghold dating back to the 11th century [32]. This is consistent with the use of horse bones, mostly found in the older strata dating back to the second half of the 10th and 11th century (see above). Horse hide products are also known from other early medieval sites, but their frequency is insignificant [50,51,52]. However, this did not result from the sentiment towards horses or a reluctance to treat their hides, which is clearly exemplified by the excerpt from The Russian Primary Chronicle (The Tale of Bygone Years) on skinning animals alive on the battlefield. The most probable reason was rather the questionable quality of horse leather. It has uneven thickness and is quite loose, except for a thicker, quality part around the horse’s rump, which was used for the production of soles, belts and pieces of military equipment, e.g., combat jackets, saddles, shields, etc. [50,51,52]. However, horse leather was of limited use for the production of clothing, as it chafes the human skin. Foal hide is a lot better for this purpose, but young specimens were mostly trained for riding [50].

The use of horse hooves (hoof capsule) and the presence of individual items made of this material were mentioned above. Due to their high collagen content, hooves were also an excellent material for the production of adhesives [50]. Although such a product is very hard to find in the archaeological material, the finds from Ostrów Tumski seem to confirm this. The above-mentioned horse remains discovered in one of the wells lacked not only metacarpal and metatarsal bones, but also most of the hooves. However, no hoof capsules were found on any of the preserved specimens, and are believed to have been deliberately removed (Figure 4a–i), probably for glue production. However, it should be added that in the case of other animals, phalanges (including distal phalanges) are usually found on skinning sites [53]. Perhaps the limited use of horse raw material for adhesive production may explain the weak participation of horse hides in tanning, and their mutual relationship translated into the number of limb skeletal elements.

Horsehair also had its application. To this day, it is used to produce brushes. Moreover, the bristle of horses and other animals was used in tanning and as a filling for pillows, mattresses, etc. [50,54]. In the Wroclaw stronghold, it was found in pits used for tanning [55]. Interestingly, horsehair often accompanied skates made of horse bones (Figure 5a) [55]. This could indicate the comprehensive processing of all available types of horse raw material in one place. Instrument strings and cords were also made of horsehair [54] and the latter was excavated in Ostrów Tumski (Figure 5b) [56]. A decorative braid described as a horsehair necklace was also found (Figure 5c) [57]. This kind of braid, sometimes interpreted as a wreath, is also known from other sites in Poland. Apart from being a decoration on its own, it is also thought to have apotropaic as well as purely practical functions, as a string for an ornament or amulet [39].

The horse’s organic remains had a number of uses in folk medicine. Drinking its blood was supposed to prevent anaemia. Feces was used to cure indigestion, while ground horse hooves and saliva were said to alleviate dermatological ailments. The latter supposedly was also a cure for constipation [58]. Drinking horse milk has been practiced since antiquity, and it is confirmed in written sources, also in relation to the early Middle Ages [40]. Archaeology, unfortunately, does not provide evidence of such practices, but it can be assumed that they took place in the Middle Ages.

### 3.5. Horse Meat Consumption

Not much can be said about the consumption value of horse meat. Nowadays, horse meat is very rarely eaten in most countries, despite its good nutritional quality. Properly prepared it is considered a delicacy, but there is strong resistance to its eating. The consumption of horse meat is noticed rather in the regions of nomadic traditions (i.e., Kazakhstan, Mongolia) [21]. It is important for our study to establish an approach to horse meat consumption in the early Middle Ages. On the one hand, the connection between horses and the sacred zone is evident, and on the other hand, malnutrition was a common problem at that time [14,19,20]. The prevailing opinion is that horses were not bred for meat, but for riding and transportation, which is confirmed by the advanced age of the horses from Ostrów Tumski and other sites. It is also indirectly indicated by the frequency and condition of the bones of other mammals. Domestic animals predominated in the stronghold of Wroclaw (94.5%), among which the most common are the bones of pigs (44%), cattle (32%), and to a lesser extent small ruminants (sheep and goats, 12.8% on average). Their number significantly exceeds the proportion of horse bone remains (3% on average, Table 1). The bones of all these species (except for horses) showed significant disintegration and numerous signs of cuts and splitting in order to butcher the carcass or obtain marrow [6,7,31]. The good condition of the horse remains from Ostrów Tumski, emphasized by archaeozoologists (the presence of mostly complete, not chopped bones, with a lack of burned bones or any other signs of culinary use), also indicates no interest in the animal’s meat [26,30,40]. However, this is not always the case. Single horse bones from the stronghold show signs of cutting and burning. In the case of limb skeletal fragments, they could have been created during the collection of raw material, e.g., for making skates. However, the presence of such traces on other parts of the skeleton should rather be interpreted as related to carcass portioning and meat consumption [7]. Such signs of horse meat consumption are also identified in other Wroclaw [8] and Central European sites [40,46,59]. It should also be added that it might have not only been consumed by humans, but also hunting dogs [24]. However, other examples confirm that it was not a common phenomenon, but rather an attempt to supplement the diet with the available type of meat (killing a wounded or lame horse). One should bear in mind extreme circumstances such as sieges, when each source of protein becomes invaluable [1,3].

## 4. Conclusions

The horse remains analyses from the Wroclaw stronghold showed the whole spectrum of the various applications. Horse bones were a common and preferred raw material for the manufacturing of certain items, such as ice skates. Other parts of the horse skeleton (splint bones, hooves) were also used, as well as horsehair and hides. Horse meat was also eaten at times. Probably only over-average-sized specimens with a greater material (and probably sentimental) value for the owner were saved. Sometimes religious considerations could also be an impediment to the widespread use of horse remains, e.g., for the production of skates, but the presence of other items made of horse bones among communities strongly associated with horses (e.g., the Balts) seems to prove that no particular importance was attached to it.

This image clearly contradicts the vision of the horse only as a mount or a sacred animal of the Slavs. However, this is only a seeming contradiction. A medieval man was practical by nature, and living conditions did not allow for wasting valuable raw materials. This is fully reflected in the studied objects. Horses, unlike other domestic animals, usually lived to old age. However, after their death, there was no hesitation in using their remains. Although the meat of old animals was no longer fit for consumption and the usefulness of their hide was limited, the bones of limbs and probably horsehair were used, and their skulls were buried under the houses as offerings. It was a symbolic connection both in the sacred and profane spheres of two culturally and emotionally close species: a human and a horse.

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
