# Peer review of "Different Approach to Horses—The Use of Equid Remains in the Early Middle Ages on the Example of Ostrów Tumski in Wroclaw"

_animals, 2020, doi:10.3390/ani10122294_

Round 1

Reviewer 1 Report

Without being a specialist in the matter shown in this article, I think it is fine but that there are aspects that must be improved. The introduction should be expanded. The methods should be expanded, and should further explain the statistics used, the tests used and specify what they are used for.

figure 1 should have a caption explaining the different colors. 

The authors could show other examples of horse exploitation in other European deposits.

in the line 213:"The most numerous were the remains of horses from the size group (c)", what is the group C?

Author Response

Figure 1 has been corrected (description in legend completed)

The introduction have been expanded. The purpose of the use of the methods has been explained, but some methods are also presented in the subsection Statistical analysis of the horse skeletal remains

Information on the 'c' size group (medium horses, 136.4-144.0 cm at the withers) can be found in line 210. The sentence in the line 213 has been modified to clarify.

Reviewer 2 Report

This work emphasizes the importance of horses life and death, for the residents of Ostrów Tumski in Wroclaw in the early Middle Ages. The goal of this study is to clarify the centrality of the horse in the daily life of the ancient inhabitants. The horse proves to be very effective not only during its life but also after its death when its bones, hair and flesh help the ancients.

Sometimes, however, the article has inaccuracies, requiring correction and focus of data collection methods and their interpretation.

Attached are some comments requiring the authors' corrections before publishing the article:

  • This work does not indicate the presence of non-horse equids on the site. Researchers must show the reader how the remains of donkeys or mules were separated from horses. There is great difficulty in identifying these species. Therefore, researchers must present their method of work.
  • Diagram 1 located during the introduction is not clear and its location does not fit the flow of article.
  • The figure is important and significant but its reference in the text is limited. The authors don't refer to the correlation between location usage (private or industrial) and the presence of equids. Is the same frequency expected in private houses vs. industrial location?
  • As in the previous paragraph: it's probable that the frequency of equids will differ between different settlements, for example farms, villages and towns.
  • Figure 2 is important and helpful and shows well the distribution of horses' skeletal parts by the frequency of their processing. It should be noted that this pattern is also seen in previous studies, where long and straight bones are preferred for tools.
  • It's correct that little evidence exists for horse meat consumption, and a main usage of horse as a pack animal. Yet, it's better to add a simple comparison of horses to edible animals like pigs or sheep. Specifically, it would be possibly helpful to compare the frequency of cut-marks on animals' bones.

It's correct that little evidence exists for horse meat consumption, and a main usage of horse as a pack animal. Yet, it's better to add a simple comparison of horses to edible animals like pigs or sheep. Specifically, it would be possibly helpful to compare the frequency of cut-marks on animals' bones.  

Generally, the article is organized as a good review about the exploitation of horse remains by the inhabitants of Ostrów Tumski in Wroclaw in early Middle Ages Poland, yet better care of the above details can help in improving the article.

Author Response

As well known, the differentiation between various types of equids raises many problems due to their morphological similarities. During the morphological sciences development many attempts for mentioned problem solving were carried out. The archaeozoological methods based on the morphometric investigations, landmarks comparison or enamel structure are known, but they give ambiguous results, have many disadvantages and weak points. In 2017, Hanot et al. ‘Identifying domestic horses, donkeys and hybrids from archaeological deposits: A 3D morphological investigation on skeletons’ introduced new method . The accessible material form Wrocław Ostrów Tumski had been excavated and elaborated since 60-ties of the 20th c., unfortunately earlier works lack of any differentiation between equids. The bone assemblage were classified as coming from horses, modern elaborations (including us), gave the same conclusions. What is typical for the majority of archaeozoological works. It is possible that in future we will be able to study this subject one more time in order to make clear equids differentiation. If reviewer have an opinion, that term ‘equid’ use instead horse is better, we will change it. The information about the lack of differentiation between types of equids was added.

The Diagram 1 was corrected and moved in text to more adequate place.

Figure 1 has been moved to the section Statistical analysis of horse skeletal remains, adding a broader comment to the concentration zone of horse bones at the site. In our opinion, it is not related to the existence of any workshop. This concentration is actually the result of better recognizing this part of the site.

It would be a very good idea to compare horse bone frequency in villages and towns. The problem is that for the analysed period (10 – 12th /13th century), the vast majority of analyses of horse bones that may be useful for this study concern other strongholds. This period is also too early to speak of a clear rural and urban subdivision. The first town location (Magdeburg Law) occurred in Złotoryja (Ger. Goldberg) before 1211.

Information on the chronology of the site has beed added to the article as it had not been properly explained before.

It was emphasized that long bones were also most often used in other positions. Their shape (straight course of bone shaft long axis) made them the most useful.

The comparison of horses to other edible mammals has been done. However for the site no detailed analyses of the frequency of occurrence of cuts on the bones of the horse and other animals were performed. It was only noted that the horse bones were usually completely preserved, while the bones of other animals were heavily fragmented.

Reviewer 3 Report

This article presents some interesting new findings on human-horse interactions in the Medieval period. The treatment of the topic is thorough. The main lacking element is a driving question behind the research. Presenting a clear question in the beginning would be helpful to guide the reader through the results. Why is it so important to study the roles of horses during this time period? I also suggest reorganization of the results and discussion sections. Separating out the findings of this study from those of others will help the reader to more clearly evaluate human-horse interactions at the site. Also, greater separation of the results from the discussion would help to gradually build up the interpretations instead of moving back and forth between data and interpretation. Please also see the line by line suggestions that I have provided. It is an interesting paper and will be a worthy contribution for publication with some reorganization and clarification of the main question of the authors.

Notes:

Simple Summary and Abstract- Could you add the question that you would like to investigate to the simple summary? Why is it important to study horses at this site? What can we learn about people during the Medieval period from this study?

Main text notes:

lines 46-47: It accompanied the man both in life and afterlife. --- Please clarify what this means

lines 50-59: Are there any types of natural taphonomic agents that may also impact bone loss?

Diagram 1: Elements/parts are not defined in relation to the colors of the bars. It would also be easier to read this if you displayed the differences from the normal pattern with statistical methods instead of these visual comparisons of the bar lengths. This diagram also looks like a result, so it should be moved to a results section rather than the introduction.

Figure 1: It is difficult to distinguish between the differently sized markers because they have fuzzy boundaries. I would suggest using three differently sized markers and better defining their boundaries. Also, please use a different shape for the “presence” marker as this will be indistinguishable from the other markers when viewed in black and white.

Line 82. Incomplete sentence: Constitute another important element of the analysis.

Line 91: clean up

Lines 91-95: This seems like another topic. Perhaps add why this will add to the methods performed before.

Line 108: The percentages do not have meaning without the NISP. Is this in a table somewhere?

Lines 103-118: Break up this paragraph by theme. You start with intra-site results and move to intra-region comparisons in the same paragraph. These need to be separated as it is confusing.

Lines 127-135: Much of this paragraph belongs in the discussion section rather than the results.

Figure 2: This reads as though the number of artifacts comes from the website.

Figre 3: This is a great figure.

Line 203: Does the withers height variation tell us anything about interactions among cities in the region? What does the diversity of horse types mean for horse trading or the scale of economic interactions that horse could have been involved in?

Line 241: fix “bones traces”

Line 271: fix “explain low share”

figure 4: Please indicate which part of the hoof was removed in the figures with an arrow. Were they cut? Also Photo (j) does not show a distal phalanx. It is a cranial fragment.

Line 283: Fix “This type of braids”

Lines 290-294: Does this information come from a historical text?

Lines 313-315. Combine this with the paragraph on blood since there is only written evidence of it.

Conclusion: Please add some interpretation of how these symbolic connections to horses could also influence how the horse-bone objects were used. If they are an important symbol in other cases, couldn’t they have had both symbolic and function value as craft items?

Author Response

The article was divided into thematic sections including the specific use of horse remains. This structure is accepted by the journal author’s instruction for manuscript preparation and in our opinion seems to be adequate. It was considered logical by the other reviewers. We think that breaking it down will result in unnecessary repetitions in the text, because the authors will refer again to the information contained in earlier chapters.

Simple Summary – the question about the role of the horses and the aim of the study has been added to this part

lines 46-47: It has been explained that horses were quite often buried with their owners in  early middle ages and were the element of some sacral practise of that times.

lines 50-59: Of course, the natural taphonomic factors can play an important role in skeletal remains preservation. The horse’s bone remains were coming from adult and healthy animals (fully ossified and well mineralized), what allow for statement that they were able to survive in good condition. The size and type of bone (small and fragile bone fragments, i.e. carpal bones, hyoid bones etc.) could be destroyed or even missed during excavation, but they were also not used for any tools manufacturing. The chemical characteristic of soil and the underground water existence can also have an impact on bones preservation (i.e. low pH can demineralize bone tissue), but such conditions were not observed in investigated site due to the bone size, pH of soil etc. The floral and faunal influence on bone assemblage (i.e. root of plants can damage bones or animals are able to drag out bones from fresh wastes pits or carcases.

Diagram 1: The colour bars description was added. We hope this make the diagram easier readable. The location of diagram in text was changed, too. If any changes are still needed, we are open for any suggestions.

Figure 1 has been corrected as indicated by the reviewer

Line 82: the word ‘statistic’ disappeared when pasting Figure 1. Consequently, the sentence no longer makes sense. Improvement is in the original text.

Lines 91-95: this paragraph has been moved to the above paragraph to make this part of the text more clear.

Line 108: The TNF, NISP and percentages estimated on the basis of accessible bone material were presented in table 1 and inserted into manuscript.

Lines 103-118: The paragraph has been broken.

Lines 127-135: As we decided to not separate chapter Results and Discussion in two, due to complete transformation of manuscript and edition difficulties, the mentioned lines stay not changed.

Figure 2: the figure description has been corrected.

Line 203: these  remarks were elaborated in the text, but with reference to the realities of the early Middle Ages.

Line 241: it has been corrected

Line 271: it has been corrected.

Figure 4 and the associated part of the text had been corrected.

Line 283: 'This type of braids' has been corrected.

Lines 290-294: yes it does, but it concerns a folk medicine so the sources are dated to 19th and 20th century.

 Lines 313-315. This paragraph was combined with the fragment form chapter above, as the Reviewer suggested.

Some remarks on the relationship between the symbolic / economic role of the horse and its use as a raw material has been added to conclusion, using selected examples from the text. However, the data is too limited to answer the question of whether the  artefacts made of horse had any symbolic function. This problem requires further study.

Round 2

Reviewer 2 Report

This work emphasizes the importance of horses and their remains, for the residents of Ostrów Tumski in Wroclaw in the early Middle Ages. In its revised version the article is worthy of publication. Most of the inaccuracies identified have been satisfactorily corrected and the article can be accepted in its new form. However, it is important that authors address the below problems:

  • This work indicate the presence of non-horse equids on the site during the review of the research methods. It might have been appropriate to highlight this reservation and not just refer to horses throughout the article, but it is acceptable..
  • Diagram 1 was rightly removed from its inaccurate place.
  • It is correct that little evidence exists for horse meat consumption, and that the horses were mainly used as a pack animal. However, it was possible to expand on the taboo on eating horse meat among various societies today and the lack of evidence for this in the past, even though this animal was sacred to the Slavs.

Generally, the article is organized as a good review about the exploitation of horse remains by the inhabitants of Ostrów Tumski in Wroclaw in early Middle Ages Poland, yet better care of the above details can help improving the article. However, this article can be published.

Author Response

It has been emphasized that the article also concerns other (non-horse) equids: title (line 2), simple summary (line 26) and introduction (lines 74-88)

The fragment about different attitudes to the consumption of horse meat has been added to the text: introduction (lines 89-101), results and discussion (lines 384-400).

Reviewer 3 Report

The author has made good additions to improve the paper. The title and the added text need some minor editing for English. Otherwise, the paper has been significantly improved. Also, Figure 4 is much clearer now, but an explanation of what the arrows indicate could be added to the figure caption.

Author Response

The title and the added text have been edited.

Explanation what the arrows indicate have been added to the figure caption.